# Evaluation of Malondialdehyde Levels, Oxidative Stress and Host–Bacteria Interactions: *Escherichia coli* and *Salmonella* Derby

**DOI:** 10.3390/cells11192989

**Published:** 2022-09-26

**Authors:** Vardan Tsaturyan, Armen Poghosyan, Michał Toczyłowski, Astghik Pepoyan

**Affiliations:** 1Military Therapy Department, Yerevan State Medical University, Yerevan 0025, Armenia; 2The International Scientific-Educational Center of NAS RA, Yerevan 0019, Armenia; 3Food Safety and Biotechnology Department, Scientific Research Institute of Food Science and Biotechnology, Armenian National Agrarian University, Yerevan 0009, Armenia; 4Scientific Research Institute of Food Science and Biotechnology, Armenian National Agrarian University, Yerevan 0009, Armenia

**Keywords:** oxidative stress, malondialdehyde, intercellular interactions, probiotic, commensal *E. coli*, host–bacteria interaction

## Abstract

Either extracts, cell-free suspensions or bacterial suspensions are used to study bacterial lipid peroxidation processes. Along with gas chromatography-mass spectrometry, liquid chromatography-mass spectrometry, and several other strategies, the thiobarbituric acid test is used for the determination of malondialdehyde (MDA) as the basis for the commercial test kits and the colorimetric detection of lipid peroxidation. The aim of the current study was to evaluate lipid peroxidation processes levels in the suspensions, extracts and culture supernatants of *Escherichia coli* and *Salmonella* Derby strains. The dependence of the formation of thiobarbituric acid-reactive substances levels in the cell extracts, the suspensions and cell-free supernatants on bacterial species, and their concentration and growth phase were revealed. The effect of bacterial concentrations on MDA formation was also found to be more pronounced in bacterial suspensions than in extracts, probably due to the dynamics of MDA release into the intercellular space. This study highlights the possible importance of MDA determination in both cell-free suspensions and extracts, as well as in bacterial suspensions to elucidate the role of lipid peroxidation processes in bacterial physiology, bacteria–host interactions, as well as in host physiology.

## 1. Introduction

The interdependence between the lipid peroxidation processes (LPP) and the composition of lipids can be considered both as a physicochemical system of regulation and as one of the normal forms of renewal of the composition of lipids of bacterial cell membranes [1,2]. However, prolonged non-enzymatic free radical oxidation of lipids leads to a sharp disruption of the physicochemical structure of membranes. This, in particular, relates to permeability, stability of lipid–protein complexes, as well as inactivation of lipid-dependent membrane-bound enzymes [3,4,5]. Detecting the dynamic variations of oxidative stress is informative for the clarification of its impact in the basic cellular processes and for its regulation [6]. One of the most popular markers for the assessment of oxidative stress is malondialdehyde (MDA) [7], the endogenous genotoxic product for both enzymatic and non-enzymatic LPP [8,9]. The different approaches are used for the sample preparation to monitor the concentrations of MDA for bacteria, including homogenates of fresh [8] and freeze-dried bacterial cultures [10] and bacterial culture broths [6]. Different mechanical pressure, acoustic, temperature and chemical methods for the extract preparation for bacteria are also used [11], and the mechanical methods are evaluated as one of the appropriate methods for oxidative stress monitoring among these.

A diverse community of large numbers of commensal bacteria from the human and animals’ mucosal and epidermal surfaces plays a crucial role in host life [12,13,14,15,16]. To attempt to understand the immunologic impact of individual commensal species within the microbiota, Brown and co-authors systematically profiled the immunologic fingerprint of commensals from the human intestinal major phyla, showing that *Bacteroidetes* and *Firmicutes* have distinct effects on intestinal immunity by differentially inducing genes’ primary and secondary responses [17]. Parallel to the species from *Actinobacteria*, *Bacteroidetes* and *Firmicutes,* the importance of gut commensal *Escherichia coli* (*E. coli*) from the Proteobacteria for the human host is described in the literature [18,19,20]. A series of our publications, mainly relating to the changes in growth of gut commensal *E. coli* in diseased people or animals, also emphasize the importance of gut commensal *E. coli* for the host’s physiology [21,22,23,24,25,26]. DNA synthesis is sensitive to changes of phospholipids, fatty acids and lipid peroxidation in the bacterial membranes [1,27].

Previously, we hypothesized that membrane interactions influenced the assessment of LPP intensity in the bacterial extracts and suspensions of *Salmonella* Derby strains K89 and K82 [8,28]. Taking into account this and all the above-mentioned information, the aim of the current study was LPP evaluation of *E. coli* G35 strains from healthy and diseased human gut microbiota, with attention on the changes in thiobarbituric acid-reactive substances (*TBARs*) levels in the bacterial extracts, suspensions and in the cell-free supernatants. In addition to our previous investigations [28], the TBARs levels of the *S.* Derby strain K134, the ultraviolet-sensitive (UV) mutant strain, were also evaluated during the current study. The main point of the research was whether the determination of malonic dialdehyde/TBARs levels in bacterial suspensions, extracts and cell-free supernatants could be important in elucidating the role of bacterial lipid peroxidation processes in bacterial physiology, bacteria–host interactions and host physiology.

## 2. Materials and Methods

### 2.1. Bacterial Strains

*E. coli* strain G35 N49, from the feces of a breast cancer patient [8,21], probioic *E. coli* strain G35 N61 from a healthy person (Passport N 01-07/89, State Research Institute of Standardization and Control of Medical Biological Preparations, Moscow, Russia) [29] and UV-mutant strain *Salmonella enterica* subspecies *enterica* serotype Derby (*S.* Derby) K134 and K95 from the microbial strain collection of the Armenian National Agrarian University were used during these investigations.

### 2.2. Bacterial Suspension/Extract Preparation

A colony of *E. coli/S.* Derby cells were inoculated in 12 mL of Luria–Bertani broth (Sigma-Aldrich, UK) and were grown at 37 °C for 2.5–3.0 h (logarithmic phase culture) and for 24 h (stationary phase culture). The bacterial suspensions with different optical densities (OD), measured spectrophotometrically (spectrophotometer SF-46 LOMO, Saint Petersburg, Russia), were prepared from the logarithmic phase/overnight cultures at OD_600_. For this, after centrifugation (5000× *g*) for 5 min of the exponential or stationary phase bacterial culture, the different optical densities (OD_600_ = 0.2–0.7) were adjusted with distilled water using the spectrophotometer.

For the preparation of bacterial extracts from the suspensions, the suspensions were vortexed with the 0.5 mm glass beads (Cat No./ID: 13116-400, QIAGEN, Milano, QIAGEN) (eight times for 20 s, under cold conditions), and the content of MDA was determined in the cell-free extracts.

### 2.3. MDA Determination 

The LPP in bacterial suspensions, extracts and cell-free supernatants was assumed by thiobarbituric acid (TBA) reaction according to Vladimirov and Archakov [30]. The incubation mixture, at a volume of 1 mL, consisted of 40 mM Tris hydrochloride buffer (pH 7.4), ammonium iron(II) sulfate hexahydrate (Sigma Aldrich, Glasgow, UK) (12 µM) and ascorbate (Sigma Aldrich, UK) (0.8 mM). The incubation was carried out at 37 °C for 0.5–2.0 h. To investigate the impact of the growth phase on the formation of TBARs levels in cell suspensions of *E. coli,* the duration of incubation was 2 h. The “TBA”-reaction was stopped by adding trichloroacetic acid (Sigma Aldrich, UK). The residue was removed by centrifugation at 3000× *g* for 5 min. The concentration of MDA was calculated per mg of protein․ The protein concentration in the cell suspension was calculated by the use of 2 N NaOH with a modification of Yakovleva and co-authors [31]. For this, 1 mL of 2 N NaOH was added to 80 mg of cells; after incubation at 37 °C for 18 h and centrifugation at 5000× *g* for 5 min, the concentration of protein was determined in the supernatant according to a Lowry assay [32]. The protein concentration in the extracts was also determined according to Lowry [32]. 

After the centrifugation of the bacterial suspension at 5000× *g* for 5 min, 0.2 mL of the supernatant was used for the determination of TBARs levels in cell-free supernatants [8,27].

MDA in *E. coli* G35 bacterial suspensions was determined both by the above mentioned method and by the instruction of the commercial kit (Lipid Peroxidation (MDA) Assay Kit (Colorimetric) (ab233471)), making it possible to compare the results.

### 2.4. Statistical Analysis

The Mann–Whitney and Student’s t tests (Excel 16) were used for statistical analyses. A probability of *p <* 0.05 was considered significant.

## 3. Results

### 3.1. *TBARs* Levels in the Bacterial Suspensions

The results for *TBARs* levels in bacterial suspensions are shown in Figure 1. The results indicate that the maximum amounts of MDA for bacteria are observed after 1.5 h. The formation of malondialdehyde is observed depending on the reaction time (0.5 h–2.0 h) at OD_600_ = 0.4 and OD_600_ = 0.7 concentrations of bacteria for *E. coli* G35 N61 cells: 95.34–120.91 μg/mg protein (OD_600_ = 0.4) and 31.1–55.1 μg/mg protein (OD_600_ = 0.7) (Figure 1). In the case of *E. coli* G35 N49 with a concentration of OD_600_ = 0.4, 118.47–150.26 μg/mg protein of MDA is formed within 0.5 h–2.0 h; at a concentration of OD_600_ = 0.7, 38.65–68.47 μg/mg protein of MDA was formed. At a concentration of OD_600_ = 0.4 of *S.* Derby K89 within 0.5 h–2.0 h, the MDA is formed with 93.47–118.54 μg/mg protein concentrations, and at a concentration of OD_600_ = 0.7, the MDA is formed in 30.39–54.02 μg/mg protein concentrations. For *S.* Derby K82, *S.* Derby K134 and *S.* Derby K95 strains, at OD_600_ = 0.4, an increase of levels of MDA (30.58 μg/mg protein to 36.95 μg/mg protein, 23.9 μg/mg protein to 47.44 μg/mg protein, and 8 μg/mg protein to 35.4 μg/mg protein) is recorded. Respectively, at OD_600_ = 0.7, an increase of MDA concentrations from 19.81μg/mg protein to 26 μg/mg protein, 15 μg/mg protein to 26.1 μg/mg protein and 21.2 μg/mg protein to 30.7 μg/mg protein is recorded. Moreover, within 2 hours, at OD_600_ = 0.4 concentration of *E. coli* G35 N49 strains, MDA is formed at 120.91 μg/mg of protein, and at OD_600_ = 0.7, 55.1 μg/mg of protein (Figure 1). A similar trend is observed for *E. coli* G35 N49 and *S.* Derby K134, and for *S.* Derby K95 at a concentration of OD_600_ = 0.7, an increase in the amount of malondialdehyde formed is observed, starting from 8 µg/mg of protein and increasing to 21.2 µg/ mg of protein (Figure 1).

### 3.2. *TBARs* Levels in the Bacterial Extracts

The results on TBARs levels in the bacterial extracts are presented in Figure 2. According to the results of Figure 2, the levels of formed MDA, depending on the species/strain of bacteria, remained relatively constant or increased in bacterial extracts of OD_600_ = 0.4 and OD_600_ = 0.7 concentrations (Figure 2). 

In addition, the levels of formed MDA increased or stayed relatively constant in the bacterial extracts depending on the duration of the TBA reaction (Figure 2). Thus, after 0.5 h of incubation, the content of MDA for the bacterial extracts (OD_600_ = 0.4) was 38.7 ± 1.93 μg/mg protein. There were no significant statistical changes in this “value” during the following incubation periods: 38.7 ± 1.93 vs. 40.05 ± 2.0, 40.41 ± 2.02 and 40.77 ± 2.04; 32.47 ± 1.62 vs. 33.64 ± 1.68, 33.94 ± 1.7 and 34.24 ± 1.71 (for the *E. coli* cells); and 22.40 ± 1.12 vs. 23.52 ± 1.18, 23.71 ± 1.19 and 23.80 ± 1.19 (for the *S.* Derby K89 cells) (*p >* 0.05) (Figure 2). In the case of *S.* Derby K82 cells (OD_600_ = 0.4), the increase of the content of MDA was detected after 1.5 hour of incubation, in comparison with that after 0.5/1 hour of incubation (Figure 2); the level remained unchanged in 2.0 h of incubation (Figure 2). In comparison with the concentration OD_600_ = 0.4, the levels of MDA increased for OD_600_ = 0.7 (all the investigated bacterial cells) (Figure 2).

### 3.3. *TBARs* Levels in the Logarithmic and Stationary Phase E. coli Cells

The results of TBARs levels for the logarithmic and stationary phase *E. coli* cells are presented in Table 1. According to these results, the concentrations of MDA formed for the bacterial suspensions of *E. coli* received from the logarithmic and stationary growth phases were different (Table 1). 

### 3.4. *TBARs* Levels for the *E. coli* G35 Cells

The results on comparative analysis of TBARs levels for the stationary phase *E. coli* G35 cells are presented in Table 2. According to these results, the concentrations of MDA formed in the bacterial suspensions, extracts and cell-free supernatant of *E. coli* G35 N61 (probiotic strain) differed from that of the *E. coli* G35 N49 strain (Table 2).

## 4. Discussion

### 4.1. *TBARs* Levels in the Bacterial Suspensions and Extracts

As expected (according to the instructions of commercial kits), the formation of maximum amounts of MDA in bacterial suspensions is observed after 2 h for all *E. coli* and *S.* Derby cells. This proves that 2 h is sufficient for a correct quantitative assessment of MDA in bacterial suspensions. According to the present study, the amount of malondialdehyde formed in bacterial suspensions, in addition to the species and strain of bacteria, as well as the duration of the reaction, also depends on the concentration of bacteria in suspensions. It is interesting to note that sometimes in the case of bacterial suspensions and more rarely for the extracts, higher concentrations of “OD_600_ = 0.7” show a lower level of MDA compared to lower concentrations of “OD_600_ = 0.4” (Figure 1 and Figure 2).

These results are in accordance with our preliminary investigations, where the possible influence of membrane interactions on assessment of LPP intensity in the bacterial extracts and suspensions of *S.* Derby were discussed [8,28]. The levels of MDA for both *E. coli* and UV sensitive mutant cells of *S.* Derby K134, detected during the current study, refer to the formula (1) for the bacterial suspensions, and formula (2) for the bacterial extracts, as previously were supposed by Pepoyan and co-authors [8,28]:d[MДA]_c_/dt = K_o_N [MДA]_o_ (S_o_ − S_k_)(1)
d[MДA]_c_/dt = 2 K_o_N [MДA]_o_ (S_o_ − S_k_^э^)(2)
where K_o_—the constant of the process of MDA release from the membrane into the intercellular space, N—the concentration of bacterial cells in the system, [MDA]_o_—the concentration of MDA, S_o_—surface of cell membrane, S_k_ and S_k_^э^—surface of cell membrane engaged in intercellular contacts for the “intact” and “destroyed” cells, and (S_o_ − S_k_)—the surface of the membrane free from intercellular contacts.

Integrating the Equation (1)/(2), for the relatively constant bacterial concentrations (S_k_ = constant and S_k_^э^ = constant), the Formula (3) can be formed: [MДA]_c_= S_o_ − S_k_/2(S_o_ − S_k_^э^)[MДA]_э_(3)

Formula (3) provides evidence of the existence of dependence between the formation of MDA in the system, and S_k_ and S_k_^э^ both for the “intact” and “destroyed” cells of *E. coli* and *S.* Derby. Consequently, at S_o_ = constant, the ratio [MДA]_c_/[MДA]_э_, determined experimentally, will be determined by S_k_ and S_k_^э^, depending on the concentrations of bacteria. In the case of extracts, one should expect a rapid increase in S_k_^э^ with a concentration and a steady-state value being reached even at lower cell concentrations than in the case of intact cells. This, perhaps, is due to the complementarity of the membranes of the intact cells, which have a spherical or ellipsoid shape. According to our previous studies, *E. coli* and *S.* Derby cells differ from each other in their morphological and physiological properties [21,25], which could be the reason for the change in the S_k_ and/or S_k_^э-^ surfaces. Bacteria in different logarithmic and stationary phases of growth also differ in their membrane and metabolic properties, which probably causes changes in the S_k_ and S_k_^e^ surfaces and, therefore, affects the formation of MDA, the end-product of oxidative stress of the same bacteria (Table 1).

Based on the above-mentioned data/discussions, the actual amount of MDA formed by the studied bacteria is observed at an OD_600_ = 0.4, when, according to the results of this study, the concentration of bacteria (intercellular interactions) does not significantly affect the release of MDA into the intercellular space (Formula (3), Figure 1 and Figure 2).

#### 4.1.1. Comparative Evaluation of *TBARs* Levels in the Bacterial Cells

Oxidative stress in the cells is determined by the predominance of levels of reactive oxygen species over antioxidant levels [33]; for example, the intracellular concentration of hydrogen peroxide under aerobic conditions increases in *E. coli* by ∼0.2 nM O_2_^−^ and ∼50 nM H_2_O_2_ [34], which can change under the influence of exogenous factors [35]. To mitigate damage caused by oxidative stress, bacteria activate various regulatory responses to stress, depending on the stressor’s nature [5]. In addition, the bacterial membrane is semi-permeable to H_2_O_2_, and H_2_O_2_ produced by one bacterium can enter and potentially harm other bacteria in the host microbiome [35]. The association of the gut microbiota with altered oxidative stress is now well established for neurodegenerative diseases [33].

As far back as the 1980s, several scientific publications indicated the association between the intestinal non-pathogenic *E. coli* cells and hosts with colorectal cancer [36,37]. Karapetyan, one of the authors of these publications, claimed that the *E. coli* G35 N49 strain predominates in the intestines of tumor patients, especially in the intestines of colorectal cancer patients [36]. The dominant *E. coli* G35 N49 strain isolated by Karapetyan from the patient’s fecal microbiota was described by him and later by his colleagues as a non-pathogenic, non-lactose-fermenting strain dominant in the intestinal microbiota of patients with colon cancer. Comparative properties of *E. coli* G35 N49 and the strain isolated from healthy human microbiota (*E. coli* G35 N61 strain) have been actively investigated, considering growth and proliferation, membrane properties [21,22,27,29], and interaction with tumor cells [38], and studies have demonstrated the differences in characteristics of predominant commensal *E. coli* strains isolated from the gut microbiota of different tumor patients [39]. In 2020, Tang and colleagues showed that, compared to healthy individuals, colorectal cancer patients harbored a lower diversity of intestinal *E. coli* isolates [40]. The authors hypothesized that “diseased” isolates suppressed the growth of healthy isolates under the nutrient-limited culture conditions [40]. Nowadays, not only an immense antioxidative and anti-inflammatory role of healthy gut microbiota is well known, but it is also known that altered gut-microbiota-mediated oxidative stress is associated with different diseases, even including neurodegenerative [41] and several skin diseases [42].

#### 4.1.2. Comparative Evaluation of TBARs Levels in the Cells of *E. coli* G35

The anti-cancer properties of *E. coli* probiotics [29,36,43,44], and the probiotics’ effectiveness in association with the pre- and post-radiation nutrition, are known [45,46]. Regarding the latter, Pepoyan and co-authors tried to explain the probiotic’s participation in the host’s free radical metabolism [45], which is possible also in the case of *E. coli* probiotics [3]. Previously, Mirzoyan and co-authors reported the differences in physico-chemical and physiological properties of E. coli G35 cells, related to the growth and cells’ membrane functions [21]. The strains of *E. coli G35* N49 (prevailed *E. coli* strain from the cancer patient) and *E. coli G35* N61, despite the same membrane potentials, were basically different in total and N,N′-dicyclohexylcarbodiimide-sensitive rates of energy-dependent transmembrane H^+^ and K^+^ transport, demonstrating a low level of H_2_ production [22]. According to current investigations, the *E. coli* G35 strains also differ significantly from each other by the LPP intensity (Table 2). Probably, the changes in MDA concentrations in the bacterial suspensions, extracts and cell-free supernatant of the *E. coli* G35 N49 affect not only the bacterial cells, but also the metabolic processes of the host organism.

#### 4.1.3. Comparative Evaluation of TBARs Levels in the UV-Sensitive Cells of *S.* Derby K134

According to Figure 1 and Figure 2, the content of MDA was different in the wild *S.* Derby strain K89, and in its UV-sensitive mutant strain *S.* Derby K134, indicating that the mutation has influence not only on the bacterial UV-resistance and on the bacterial growth [25], but also on the dynamics of LPP of the cells (118.54 ± 5.93 μg/mg protein vs. 47.44 ± 2.37 μg/mg protein (for suspension), and 26.30 ± 1.32 μg/mg protein vs. 23.80 ± 1.19 μg/mg protein (for extract)). The differences in LPP formed in *S.* Derby K134 and *S.* Derby K95 (plasmid-free derivative of K134 [8]) might be explained by the effect of the R-plasmid on host membrane integrity [8]. As in the case of *E. coli* G35, we hypothesize that the differences in TBARs levels in *S.* Derby suspensions may play an “environmentally” important role for the potential bacterial host.

Thus, despite the commercial kits for the assessment of malondialdehyde formation, mainly determining the amounts of malondialdehyde formed in human (also rat and mouse) serum and plasma, as well as the amounts of malondialdehyde formed in various tissue homogenates, these kits can also be easily used to assess malondialdehyde formation in bacterial extracts, suspensions, and supernatants. However, the dependence of TBARs levels in the cell extracts, the suspensions and cell-free supernatants on bacterial species, their concentration and the growth phase were revealed during the current investigations. The effect of bacterial concentrations on MDA formation was also found to be more pronounced in bacterial suspensions than in extracts, probably due to the dynamics of MDA release into the intercellular space.

On the other hand, interestingly, the comparative assessment of TBARs levels in the cell extracts, the suspensions and cell-free supernatants in one of the important representatives of intestinal microbiota, *E. coli*, revealed differences between the probiotic and the “diseased” strains (Table 2); meanwhile, usually, either extracts, cell-free suspensions or bacterial suspensions are used to evaluate bacterial lipid peroxidation processes, which may lead to incorrect discussions and conclusions. 

## 5. Conclusions

Nowadays, it is known that, in addition to commensal bacteria [47], “non-pleasant”/pathogenic microbes [21,48] can also live inside hosts without causing noticeable diseases. Along with adhesion processes, which are mainly determined by the bacterial membrane structures and are important in bacteria–host interactions, reactive oxygen species and nitrogen species are also actively discussed by researchers as a defensive tool against pathogens [49,50]. The bacterial membrane is semipermeable to H_2_O_2_, and H_2_O_2_ formed by a microbe can be destructive to the host microbiome [35]. The different approaches are used for sample preparation to monitor the concentrations of MDA for bacteria, including homogenates of fresh [8] and freeze-dried bacterial cultures [10] and bacterial culture broths [6]. Current investigations revealed levels of thiobarbituric acid-reactive substances in cell extracts, suspensions and cell-free supernatants for *S.* Derby and *E. coli* strains. This study highlights the importance of simultaneous assessment of oxidative processes in bacterial extracts, suspensions and culture liquids to elucidate the role of lipid peroxidation processes in bacterial physiology, bacteria–host interactions, as well as in host physiology. Logarithmic/stationary growth phases of bacteria might also be important in such evaluation processes. However, further investigations are needed to specifically define the role of the formed MDA in cell-free supernatants or suspensions or extracts in the bacteria–host interaction.

## Figures and Tables

**Figure 1 cells-11-02989-f001:**
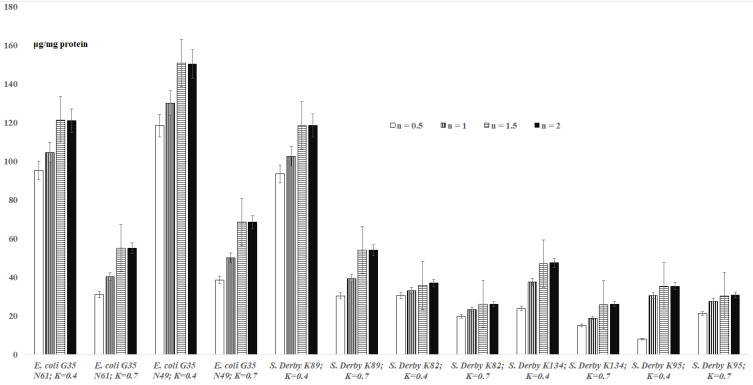
Levels of TBARs formed during lipid peroxidation processes in bacterial cells (suspensions; overage ± standard error) depending on the incubation time (n, hour); *K*—“OD_600_”.

**Figure 2 cells-11-02989-f002:**
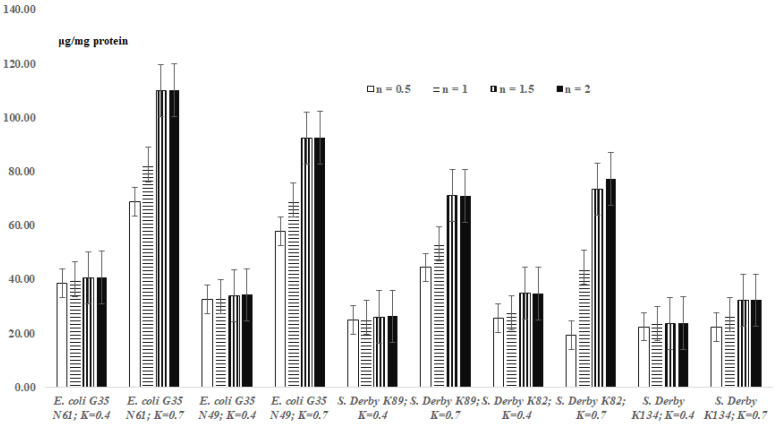
Levels of malondialdehyde formed during lipid peroxidation processes in bacterial extracts (overage ± standard error) depending on the incubation time (n, hour); *K*—“OD_600_”.

**Table 1 cells-11-02989-t001:** Impact of growth phase on thiobarbituric acid-reactive substances (TBARs) levels * in cell suspension of *Escherichia coli*, μg/mg protein (overage ± standard error, incubation time: 2 h).

Strains	Logarithmic Phase of Growth		Stationary Phase of Growth		
	OD_600_ = 0.1	OD_600_ = 0.2	OD_600_ = 0.2	OD_600_ = 0.4	OD_600_ = 0.7
*E. coli G35* N61	143.36	55.58 ± 2.78	37.06 ± 1.85 *p* * < 0.05	121.3 ± 2.4	55.02
*E. coli G35* N49	178.15	69.07 ± 3.45	46.05 ± 2.3 *p* * < 0.05	150.74 ± 1.75	68.37

* NADPH-dependent lipid peroxidation. *p* < 0.05 was considered significant (comparison of the levels of MDA in OD_600_ = 0.2).

**Table 2 cells-11-02989-t002:** Shifts in thiobarbituric acid-reactive substances (*TBARs*) levels (μg/mg protein) of *Escherichia coli* G35 cells (overage ± standard error, incubation time: 2 h).

TBARs Levels	Strains	
	*E. coli* G35 N61	*E. coli* G35 N49
Suspension	121.3 ± 2.4 (75.55) *	150.74 ± 1.75 (93.4) * *p* < 0.05
Extract	40.72 ± 1.05	34.2 ± 1.23 *p* < 0.05
Cell-free supernatant	0.625 ± 0.08	0.171 ± 0.06 *p* < 0.05

*p* < 0.05 was considered significant (comparison of the strains (OD_600_ = 0.4)). * The data received by the use of ՞ab233471՞ (the average of three experiments).

## Data Availability

The data that support the findings of this study are available on request from the author.

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
