# Peer review of "Evaluation of Malondialdehyde Levels, Oxidative Stress and Host–Bacteria Interactions: Escherichia coli and Salmonella Derby"

_cells, 2022, doi:10.3390/cells11192989_

Round 1

Reviewer 1 Report (Previous Reviewer 2)

Dear the Editor

The revised manuscript seemed significant improvement, in particular Fig. 1.

Reviewer 2 Report (Previous Reviewer 1)

The resubmitted manuscript has been greatly improved, and all the comments have been addressed. In this form it can be accepted for publication.

This manuscript is a resubmission of an earlier submission. The following is a list of the peer review reports and author responses from that submission.

Round 1

Reviewer 1 Report

The manuscript is well prepared, however it suffers severe drawback which need correction before resubmission including:

1. The applied assay measures so called TBARs not only MDA. I suggest correction of "MDA level" to "TBARs level" throughtout the manuscript.

2. Change x to symbol in Methods

3. Add protein determination in Methods

4. Add details for equipment used (model, supplier)

5. Remove word "statistically" since significant means actually statistically significant

6. Figure 1 - enlarge axes legends (non-readable)

7. What do the vertical bars show on charts? SD or SE?

8. Provide "n" in table legends/figure captions.

9. Remove repeated numeric data from text when shown in tables. 

10. "p" stands for "empirical p level"

11. In Discussion keep latin names italic.

12. Conclusions have to be rewritten, since the main aim of the stady was not the evaluation of comertial kits for MDA determination...I hope.

Reviewer 2 Report

Dear the Editor

Tsatuyan T et al reported that the levels of MDA in bacterial cells. These authors chose several strains and culture conditions in logarithmic and stationary phases for MDA assay. Results seemedf to be reproducible based on the error bar found in Figures.

Major issues:

1) Discussion appeared to be unrelated to what these authors reported in this manuscript.

Minor issues:

1) Letters and digits in Fig. 1 were too small.